# Dietary Fiber in Inflammatory Bowel Disease: Are We Ready to Change the Paradigm?

**DOI:** 10.3390/nu16081108

**Published:** 2024-04-10

**Authors:** Laura Loy, Lorenzo Petronio, Giacomo Marcozzi, Cristina Bezzio, Alessandro Armuzzi

**Affiliations:** 1IBD Unit, IRCCS Humanitas Research Hospital, Via Manzoni 56, Rozzano, 20089 Milan, Italy; laura.loy@humanitas.it (L.L.); cristina.bezzio@hunimed.eu (C.B.); 2Department of Biomedical Sciences, Humanitas University, Via Rita Levi Montalcini 4, Pieve Emanuele, 20072 Milan, Italy; lorenzo.petronio@humanitas.it (L.P.); giacomo.marcozzi@humanitas.it (G.M.)

**Keywords:** inflammatory bowel disease, dietary fiber, intrinsic fibers, prebiotic, nutrition

## Abstract

Accumulating evidence from pre-clinical and clinical studies demonstrate the benefit of dietary fibers for inflammatory bowel disease (IBD). However, the majority of patients avoid or limit their consumption to manage their symptoms during the active and remission phases, although limited research supports these long-term dietary habits. Although recent evidence-based dietary guidelines highlight the importance of promoting an adequate intake of dietary fiber in IBD patients, intervention trials have not yet clearly clarified the quality and quantity of dietary fiber that should be consumed to be equally tolerated by and provide benefit for patients with IBD. This narrative review describes dietary fibers and their characteristics, analyzes the real-word studies on the impact of dietary fiber consumption in IBD in different clinical settings, and concludes with potential future directions in fiber research, focusing on the real-world needs of characterizing the consumption of fiber-rich foods and promoting their adequate intake.

## 1. Introduction

The study of the impact of dietary fibers (DF) on gut health and IBD has grown over the last decade, with accumulating evidence showing that dietary fiber can enhance health-related quality of life, improve IBD symptoms, balance inflammation, and alter the gut microbiome [1,2,3,4]. However, although a correct amount of dietary fiber is now encouraged by recent international guidelines for consumption during the remission phase for both diseases [5,6,7], and the paradigm to avoid fiber intake in CD patients with the stricturing phenotype is also changing toward an inclusion of soluble fiber with adapted texture, in IBD patients, fiber intake is inadequate compared to national fiber guidelines, and the amount of fiber in their diet is lower compared to healthy populations [8,9].

At present, the health professionals who give nutritional advice to IBD patients have to face a knowledge gap about the quantity and quality of dietary fibers that should be consumed to be equally tolerated by and provide benefit to patients with IBD during the active and remission phases [10,11]. Additionally, emerging evidence shows that individual fiber tolerance as well as the metabolic degradation resulting in production of short-chain fat acids (SFCA) is strictly dependent on the host microbiome, thus requiring a more personalized nutritional approach to modify the consumption of fiber-rich foods [12,13].

Although different dietary models have been proposed for IBD patients, characterized by a different content of allowed and prohibited fiber-rich foods (Mediterranean diet, plant-based diets, specific carbohydrates diet, anti-inflammatory diet, etc.), very few interventional studies have been conducted in adult IBD patients to characterize the impact of dietary fiber from food sources on disease outcomes (including the impact on the gut microbiome). 

The aim of this narrative review is to describe the current status of nutritional recommendations about dietary fiber in IBD, to characterize the dietary models most commonly discussed in IBD research studies in different clinical settings, to analyze the missing clinical needs for nutritional intervention with natural sources of fiber (and not with fiber supplements), and to discuss the actual knowledge gap and potential future directions in fiber research, focusing on real-world needs to characterize the consumption of fiber-rich foods and promote their adequate intake.

## 2. Current View of Dietary Fiber

Dietary fibers (DFs) are non-digestible carbohydrate polymers that are completely or partially fermented in the large colon [14]. National authorities of specific countries such as Europe, Canada, Australia, New Zealand, Brazil, and China have decided to add this category (including non-digestible carbohydrate polymers with ten or more monomeric units) as well as non-digestible carbohydrates with three to nine monomeric units [15] to their nutritional recommendations. 

DFs can be categorized according to chemical composition, dietary origin, and physicochemical properties such as water solubility, viscosity, and fermentability [16]. 

The physicochemical characteristics of dietary fiber (degree of polymerization, solubility, viscosity, and accessibility) are the main factors influencing its fermentability [17,18], which also depends on the diversity and metabolic activity of intestinal microbiota [19,20]. The final bioproducts of fiber fermentation are short-chain fatty acids (SCFAs), with recognized beneficial roles for the gastro-intestinal tract [21].

While all dietary fibers, even the insoluble cellulose and lignins, are broadly subjected to bacterial fermentation, the term “prebiotic” can be used only for the dietary fiber described as “a substrate that is selectively utilized by host microorganisms conferring a health benefit”, as defined by the International Scientific Association of Probiotics and Prebiotics (ISAPP) [22]. Thus, prebiotics serve as a nutrient source for a specific range of genera and/or species (notably Bifidobacterium and Lactobacillus) able to selectively metabolize the prebiotic substrate, thus promoting the selective growth of beneficial bacteria producers of SFCAs [23].

The term “microbiota-accessible carbohydrates” (MACs) overcomes the previous classifications and their limitations, as it refers to dietary carbohydrates that resist breakdown and absorption by the host and are metabolically available to gut microbes. Dietary MACs can be derived from diverse sources such as plants, animal tissue, or microbes in food, but they need to undergo metabolism by the microbiota, thus encompassing the term “prebiotic fibers” [24].

It is important to note that the fibers present in different types of plants will also have variable chemical compositions as well as physicochemical properties (Figure 1). For example, pectin is more abundant in fruits and some types of vegetables, and β-glucans are found in cereals. Starchy foods that contain resistant starch include pulses, cereals, and potatoes. Thus, diets rich in plant-based foods with different types of dietary fibers exert different physiological effects in the gastrointestinal tract and support a more diverse microbiota composition [25,26].

## 3. Suboptimal Fiber Intakes in Adult IBD Patients: The Patient’s Perspective

DFs intake in adult IBD patients is suboptimal when compared to that of healthy individuals and the nutritional guidelines. In a recent systematic review of 26 studies with 4164 participants with IBD, diet adequacy was assessed by comparing usual intakes with dietary recommendations and/or the intakes of healthy controls. It was demonstrated that individuals with IBD were found to eat less total fiber than healthy controls and that fiber intakes were inadequate compared to that recommended national dietary guidelines, regardless of disease activity [9].

Several underlying reasons may be considered to understand the factors influencing fiber intake in patients with IBD, for instance, dietary beliefs, self-guided elimination diets, advice from health professionals, or information from the Internet [27].

A common patient belief is that diet could be a trigger factor for IBD relapses. For this reason, some products are completely avoided or consumed less to prevent disease relapses, although such an approach is not recommended by specialists, with fiber-rich foods like raw fruits and vegetables (particularly cruciferous vegetables) and legumes being one of the most relevant food categories [28,29,30,31].

Additionally, about 40% of patients with IBD in remission suffer from concurrent irritable bowel syndrome (IBS) [32], and a low-FODMAPs (fermentable oligo-, di-, and mono-saccharides and polyols) diet may be habitually followed to mitigate functional gastrointestinal symptoms [33,34]. However, this is often without dietetic support, leading to unnecessary dietary restriction and reduced intake of fiber, particularly prebiotic short-chain fibers (oligosaccharides).

More importantly, dietary recommendations to IBD patients can be highly variable and often contradictory, with the risk of unnecessary and prolonged food exclusions and harmful dietary manipulation.

Health professionals such as gastroenterologists, IBD nurses, and specialized dieticians are the most common sources of dietary advice or nutritional recommendations for IBD patients [30,35,36,37]. The risk of receiving dietary advice not aligned with recent international recommendations about fiber seems to be very high since many patients with IBD are broadly advised by clinicians to follow a low-fiber or low-residue diet (~10 g/day fiber) [38,39]. This advice is based on historical recommendations that a “low-residue” diet is useful for short-term control of gastrointestinal symptoms and to reduce stool output [40,41]. However, in many instances, patients are not advised to re-introduce fibers, which can result in prolonged unnecessary fiber restriction [42]. 

Also, the Internet is often used by IBD patients to obtain nutritional advice, with the risk of receiving dietary recommendations that are not in line with the newest dietary evidences. Hou et al. [27] identified 32 websites discussing recommendations for fiber intake, with 72% of these recommending avoidance of high-fiber diets or foods. 

Furthermore, health professionals giving nutritional advice should consider that an inadequate fiber intake is a matter of concern also in the healthy population compared with respective national guidelines, and a nutritional intervention in IBD patients should also require a dietary education in this specific area [43,44,45,46].

## 4. Nutritional Recommendations in IBD about Fiber Intake

Over time, our growing comprehension of the significance of fiber for gastrointestinal health has highlighted the importance of IBD patients achieving an adequate dietary intake as well [4,47]. Although research studies on the role of fiber in the prevention and treatment of IBD in humans are limited [48], existing intervention trials have shown that dietary fibers can improve IBD symptoms [49], enhance health-related quality of life [50], balance inflammation [47], modulate immune responses [51], and help mitigate dysbiosis by restoring the gut microbiome [52]. Dietary fibers mediate these beneficial effects though byproducts of dietary fermentation of MAC, principally SCFA [53,54], and also by preserving the mucus layer barrier by promoting mucus synthesis and secretion [16,55].

In light of the accumulating evidence of the beneficial effect of dietary fibers in IBD, the growing need to produce dietary recommendations for these patients has prompted several international and national IBD societies and organizations to develop nutrition guidelines for these patients according to disease activity and phenotype, with a new point of view about the consumption of fiber-rich foods. 

Globally, recommendations about fiber intake in IBD patients from the International Organization for the Study of Inflammatory Bowel Diseases (IOIBD) in 2020 [5], from the European Society for Clinical Nutrition and Metabolism (ESPEN) practical guidelines on Clinical Nutrition in IBD in 2023 [6], as well as from the British Dietetic Association consensus published in 2022 [7] are consistent with general dietary guidelines for the population. 

Although evidence from RCTs do not support recommendation of any specific dietary intervention with a whole-food diet as a primary treatment to induce or maintain disease remission in active disease, sound nutritional advice should suggest that IBD patients include a sufficient quantity of vegetables and fruits in their diet and encourage them to incorporate complex carbohydrates while restricting the consumption of simple sugars. However, an exception could apply to individuals with CD and UC who have functional bowel disorders without active inflammation, where a reduced FODMAP intake could be suggested [56,57,58] with nutritional supervision only for a short-term period; a gradual and personalized reintroduction driven by tolerance and based on quality and quantity of FODMAP is necessary for the long-term management of functional symptoms to avoid malnutrition associated with micronutrient deficiencies. 

Another exception to consumption of DFs is made for individuals with symptomatic or significant stricturing CD. This subgroup of patients could benefit from personalized nutritional support to restrict their intake of the insoluble fraction of dietary fiber and include foods rich in soluble fibers when consumed with adequate hydration. 

Similarly, the practical guidelines on Clinical Nutrition in IBD by ESPEN place emphasis on food consistency for patients with symptomatic small bowel strictures, suggesting a diet with “adapted textures” such as soft, cooked, and peeled vegetables and pureed, soft, or peeled fruits in a smoothie [6]. 

However, while the restriction of insoluble fibers could be logical for patients with stricturing CD, there are no substantial data to substantiate this practice in asymptomatic patients. A recent systematic review concluded that there is no valid justification for individuals with CD to limit their fiber intake [59]. Even though there is no definitive consensus on the ideal quantity, type, and preparation of fiber in the diet for patients with IBD, the gastrointestinal health community is gradually shifting away from the broad recommendation to avoid fiber.

## 5. Fiber Intake in IBD Dietary Models

Several studies aiming to evaluate the impact of a nutritional intervention with dietary fiber have been conducted in adult IBD patients, both in the active and remission phase. However, well-designed interventional and observational studies using a food–dietary fiber approach are very limited, while clinical and pre-clinical studies are shifting their efforts in study protocols with single fiber or fiber extracts (which do not reflect the complexity of a normal diet and the variety of fiber-rich foods) (Table 1).

In CD patients (with active or inactive disease but without stricturing pattern), different nutritional interventions with fiber-rich foods have been designed. Both Levenstein and Chiba designed a nutritional intervention to encourage consumption of fiber-rich foods like legumes, whole grains, whole fruits, and vegetables compared to a control group with an omnivore diet [60] or a low-residue diet [61]. Results from Levenstein’s group showed that there was no statistically significant difference in the outcomes between the two dietary groups, including symptoms, nutritional status, need for hospitalization, need for surgery, new complications, or postoperative recurrence. Chiba et al. demonstrated a higher rate of clinical remission and a lower rate of disease relapse in the semi-vegetarian diet (SVD) group at 1- and 2-year follow-ups compared to the omnivore diet (OD) group [60].

Even when the diet intervention group did not differ from controls in consumption of fruit and vegetables (encouraged in both groups), an increased amount compared to the national average was equally tolerated in CD patients, both in active and remission phases, in addition to their usual treatments [62].

Similarly, the multicenter RCT carried out by Lewis and colleagues showed that an increase amount of dietary fiber from vegetables and fruits compared to baseline was equally well tolerated in a group of CD patients with mild to moderate symptoms who were following a Mediterranean diet (MD) or a specific carbohydrate diet (SCD). The similarities between the two dietary patterns, both incorporating fresh fruits and vegetables, could explain the lack of difference between the two dietary groups in achieving symptomatic remission, clinical remission, and reduction in biomarkers of inflammation but with a greater ease of adherence to MD [63].

The nutritional intervention proposed in CD patients in stable clinical remission by Brotherton et al. compared the impact on gastro-intestinal symptoms and quality of life of a dietary intervention targeting an increased amount of dietary fiber for subjects by eating a half cup of wheat bran cereal per day (supplied by study teams) and in giving them general instructions for reducing sugar intake compared to general dietary instruction on gastro-intestinal function by avoiding whole grain, dairy products, and spicy foods. The authors demonstrated that consuming this high-DFs and low-refined-carbohydrate diet intervention was feasible in CD patients during a four-week RCT, with no adverse effects associated with this diet, which was shown to improve the GI function and quality of life [50].

Similar nutritional interventions have been conducted in also in UC patients (with quiescent or mild active disease or in stable remission).

In the open multicenter trial conducted by Hallert et al. [64], a daily amount of 60 g of oat bran (equivalent to 20 g of dietary fiber) was achieved for each patient enrolled in the diet group with four slices of oat bran-enriched bread and 37 mL of oat bran (suspended in water, juice, or yogurt), without any changes to the ongoing medical therapy. Following the oat bran intervention over a 12-week period, no patients exhibited signs of colitis relapse; furthermore, the fecal butyrate concentration increased by 36% at 4 weeks, while other fecal SCFA levels remained unchanged. However, the significant improvement of gastro-intestinal complaints such as abdominal pain and reflux obtained at week 12 (end of dietary intervention) was cited at week 24. The same authors were aware that the small sample size (22 patients) and the study’s limited duration (spanning only 12 weeks) constrained the ability to draw general conclusions, necessitating additional long-term investigations. James et al. conducted a two-period crossover study (two weeks on each arm with a two-week washout period) in UC-inactive patients and healthy controls randomly assigned either “high-resistant starch (RS)/wheat-bran (WB)” foods containing 12 g WB and 15 g RS fiber per day, or “low WB/RS” foods (2–5 g WB and 2–5 g RS fiber per day) [65]. The dietary source of WB (about 45% fiber) was a combination of processed twig cereal and unprocessed WB, while the RS (types 1 and 2) was coarsely ground high-amylose maize (with about 30% RS content). The different dietary target was achieved among the two groups by adding WB and RS to daily consumed foods like cereal, bread, and muffins produced in a commercial kitchen. A diet high in WB and RS was well tolerated and tended to normalize gut transit, but it had no impact on the production of fecal SCFA. Interestingly, the fecal microbiota analysis from the UC cohort showed an increased diversity within Clostridium cluster XIVa and a lower proportion of *Akkermansia muciniphila* compared to controls, which could have an impact in reduced ability to ferment dietary fibers.

However, the results from other two studies with a more complex dietary intervention, where each specific macronutrient of the diet was specifically pre-defined and balanced, seems to suggest that there is a synergic effect in adopting a healthier dietary pattern focused not only on dietary fiber. 

In the randomized, parallel-group, cross-over study conducted by Fritsch et al. [66] in 17 patients with UC in remission or with a mild disease, the tested diet was a low-fat, high-fiber diet (LFD), which provided only 10% of calories from fat, while the control diet was an improved standard American diet (iSAD), with 35–40% of calories from fat (but also higher quantities of fruits, vegetables, and fiber than a typical SAD). Though both provided diets exhibited a significant increase in fiber content and servings of fruits and vegetables compared to the participants’ baseline diets, and both diets contributed to an enhanced quality of life (QoL), only the LDF specifically resulted in a noteworthy reduction in serum amyloid A and showed a tendency towards decreased C-reactive protein (CRP), an effect not observed in the iSAD group. This implies that the combination of a low-fat diet and high-fiber intake offers additional clinical benefits beyond the recommended increase in fruit and vegetable consumption in the iSAD group.

In the randomized controlled pilot trial conducted by Keshteli and colleagues [67], a diet based on dietary components with proven anti-inflammatory properties (anti-inflammatory diet—AID) and characterized by an increased intake of antioxidants, dietary fibers, probiotics, and *n* − 3 polyunsaturated fatty acids (PUFAs) and a decreased intake of red meat, processed meat, and added sugar was compared to a diet based on the dietary recommendations of Canada’s Food Guide (CFG), primarily designed to assist Canadians in adopting a healthy, well-balanced diet. In this unpowered study, the clinical relapse rate did not differ between the two dietary interventions, although a significantly greater subclinical response (fecal calprotectin < 150 μg/g at the endpoint) was only achieved in AID patients and not CFG patients. Nevertheless, in the AID group, the decreased FCP levels from the baseline to the end of the study were not correlated with fiber intake but with decreased consumption of cured meat or saturated fatty acids and increased consumption of yogurt, seafood, or poultry. 

Globally, despite differences among studies (length of the nutritional intervention and methods used to assess tolerance and adherence to the dietary intervention and to evaluate the dietary fiber amount pre and post-intervention), in all food-based fiber intervention studies, there were high rates of remission maintenance as well as improved clinical outcomes and symptomology in the intervention groups with a high intake of fiber. Additionally, the diets were all well tolerated.
nutrients-16-01108-t001_Table 1Table 1Results from RCT and prospective interventional studies with a food-dietary intervention in IBD.InterventionDurationStudy TypeDiseaseParticipantsToleranceKey Clinical OutcomesReferenceRemissionSVD vs. OD SVD: 32.4 g/dayUp to 2 yearsProspective intervention studyCD*n* = 16 on SVD and *n* = 6 on OD (median age 26.5; range 19–77 years) 22.7% of patients underwent surgeryNo adverse effects on SVD100% remission maintenance on SVD after 1 year and 92% after 2 years vs. 67% and 25%, respectively on the OD. Cumulative disease relapse rates were significantly lower in SVD vs. OD after 2 years[60]FRD vs SD FRD:33.4+ 1.8 g/day4.3 yearsProspective intervention studyCD*n* = 32 on FRD *n* = 32 on SD 28% of patients of each group underwent surgeryWell-toleratedHigher hospitalization rate in the group that received the SD than in the experimental group (34 vs. 11 respectively). Decrease in the mean number of days of hospitalization for the experimental group (6 vs. 15 days)[62]Wheat bran cereals and limited amount of refined carbohydrates vs. Control diet Wheat bran portion = half a cup/day4 weeksRandomized single-blinded controlled trialCD*n* = 22 in the experimental group *n* = 22 in the control group (mean age 40; range 18–64 years)Well-toleratedGood patient compliance and tolerance of the experimental diet and amelioration of symptoms compared to the control group[50]LRD vs. SID LRD = 8.1 portions/week (3 g/day) SID = 26.6 portions/week (13 g/day)29 monthsRandomized controlled trialCD*n* = 36 on LRD (mean age 38) *n* = 35 on SID (mean age 42) 41.6% of patients on LRD and 31.4% of patients on SID underwent surgery19 patients on SID reported diarrhea and abdominal painNo statistical differences between the two groups, including symptoms, need for hospitalization, need for surgery, new complications, nutritional status, or postoperative recurrence[61]Low RS/WB (2–5 g RS and 2–5 g WB fiber/day) vs. High RS/WB (15 g RS and 12 g WB fiber/day)2 weeks + 2-week washout periodTwo-period crossover studyInactive UC*n* = 19 with UC (mean age 38; range 18–72 years) *n* = 10 controls (mean age 41; range 26–66 years)Well-toleratedIn UC patients, a diet high in RS and WB tended to normalize the gut transit, but did not increase the proportions of fermented carbohydrates or the production of faecal short-chain fatty acids[65]SCD vs. MD12 weeksRandomized controlled trialCD*n* = 101 on SCD (mean age= 36; range 27–46 years) *n* = 96 on MD (mean age 37; range 29.5–53 years) 28.7% of patients on SCD and 35.4% of patients on MD underwent surgeryAbdominal pain was reported by 2 participants in both arms of the trial in the first 6 weeksMD with its greater ease of adherence and additional health benefits, may be a more favorable choice compared to the SCD for the majority of patients with CD experiencing mild to moderate symptoms[63]60 g of oat bran (20 g/day DF) daily3 monthsProspective intervention studyUC*n* = 19 consuming 60 g of oat bran daily *n* = 10 controls (mean age 43.5; range 20–77 years)Well-toleratedNo signs of disease relapse for both groups. Significant improvement in GI symptoms (abdominal pain and reflux) in the oat bran group. Controls had an increase in reflux[64]LFD vs iSAD4 weeks with 2-week washoutRandomized cross over studyUC- remissive and active disease*n* = 17 (median age 41.7 years)Both diets were well-toleratedAll patients remained in remission during the study. Both diets improved QoL. Serum amyloid A significantly decreased in LFD but not in iSAD group. The trend was towards a decrease in CRP in LFD group[66]AID vs CFG AID: 22.28 ± 6.7 g/day CFG: 22.3 ± 8.3 g/day6 monthsRandomized controlled trialUC*n* = 26 on AID (mean: age 36.5; range 30–55.5 years) *n* = 27 on CFG (mean age 43; range 25–54 years)Well-toleratedNo difference in clinical relapse rate between the two dietary interventions, despite only patients on AID had a general decline in faecal calprotectin and a significantly greater subclinical response compared to patients following the CFG[67]SVD, Semi-vegetarian diet; OD, Omnivore diet; FRD, Fiber-rich diet; SD, Standard diet; LRD, Low-residue diet; SID, Standard italian diet; RS/WB, Resistant starch/Wheat-bran; SCD, Specific carbohydrate diet; MD, Mediterranean diet; LFD, Low-fat, high-fiber diet; iSAD, Improved Standard American diet; AID, Anti-inflammatory diet; CFG, Canada’s food guide; CD, Crohn’s disease; UC, Ulcerative colitis; QoL, Quality of Life; CRP, C-reactive protein.

## 6. Intrinsic Fibers vs. Isolated, Single Fiber or Fiber Extract

In conjunction with rising interest in dietary fiber intervention, numerous pre-clinical and clinical studies have delved into single fiber or fiber extracts, which lie beyond the scope of this review [68,69]. In a recent systematic review [70], eight RCT studies (seven involving UC patients and one involving CD patients) were examined, focusing on five dietary fiber supplements (wheat bran, oat bran, fructans, psyllium, and germinated barley foodstuff).

Nevertheless, a healthy diet aiming to achieve a sufficient fiber intake from a variety of plant-based sources (such as vegetables, fruits, nuts, seeds, legumes, and grains) includes whole foods with a complex three-dimensional plant cell matrix termed “intrinsic fibers” [71].

The composition of the plant food cell matrix, alongside food processing and digestion, influences how the microbiota access individual fibers, impacting both digestion and fermentation patterns [72].

While dietary fiber supplements may be beneficial in attaining adequate fiber intake in IBD patients, their clinical application requires careful long-term considerations. It is not negligible that a dietary intervention providing a correct amount of fiber from foods yields more favorable microbiome outcomes compared to low-fiber diets and fiber supplements [71,73,74].

## 7. Dietary Fiber in IBD: What Are We Missing to Change the Paradigm?

While recent guidelines have shifted the paradigm about fiber intake in IBD patients, health professionals still have several unmet needs concerning the dietary fiber of IBD patients, with both pre-clinical and clinical concerns needing to be addressed.

Future human RTCs are urgently needed to clarify more accurately the type and amount of fiber tolerated in active and remission phases as well as the contribution of dietary fibers to induce and maintain remission in IBD patients in different clinical settings. Such studies need to be adequately powered, testing the efficacy and tolerance of specific diets with objective measures (clinical, endoscopic, and nutritional) at reasonable time-point assessments to ensure the reliability of the outcomes.

Furthermore, recent investigations have also highlighted that the response to dietary fiber in IBD patients is variable [65,75,76], depending on inter-individual heterogeneity in the baseline gut microbial community composition, which is influenced by dietary intake patterns [52,77,78]. Preliminary results from both pre-clinical and clinical studies have shown the favorable direct and indirect effects of specific components of fiber on different features linked to IBD pathogenesis; for example, FOS are able to promote the growth of fecal Bifidobacteria, and RS can increase the production of SCFA and reduce the presence of harmful bacteria, while inulin induces indirect production of SCFAs through its positive effect on the intestinal microbiota by increasing Bifidobacteria. One limitation of these available studies is the scarcity of human studies on the effects of specific components of dietary fiber on particular bacterial strains in the intestinal microflora as well as on the effects of individual components during exacerbation and remission in patients with IBD [79,80,81]. Although the microbiota composition [65] and SFCAs production [64,65] were analyzed only in two of the above-mentioned nutritional IBD studies and with preliminary and conflicting results, there is a reasonable basis for linking a specific dietary intervention focused on dietary fiber with an increased amount of SFCAs and specific changes in the microbiota. 

Also considering the lack of clinical evidence directly implicating SCFAs in the resolution of inflammation or mucosal healing [49,82,83], large-scale prospective, longitudinal studies with characterization of gut microbiota composition and functions and with evaluations of SFCA (final byproducts of fiber fermentation) are strongly required to link dietary fibers, the microbiome, and SFCA’s production to specific favorable outcomes and objective measures (especially the reduction of inflammation through fecal calprotectin or endoscopic assessment). 

Given the nutritional challenges anticipated in the future as we shift towards personalized nutrition, it is crucial to critically examine our existing knowledge gap and limitations in nutritional assessments. This is especially relevant when assessing dietary fiber intake.

The first concern is about the nutritional instrument used to evaluate the amount of dietary fiber [9]. In RCTs and observational trials, a baseline dietary fiber consumption is advisable to make comparable the intervention group and the control group in order to avoid a selection bias; in single-arm, prospective studies, the amount of fiber should be equally evaluated to compare baseline consumption and final intake at the end the trial. Indeed, only these evaluations will allow us to determine if the different fiber intakes given by the nutritional intervention are correlated with pre-specified outcomes during different time-point assessments. 

The food-frequency questionnaire (FFQ) is the most frequent tool used in observational trials, followed by the 3-day food diary. In interventional trials and single-arm trials, we see more often a dietary evaluation through 24- to 48-h dietary recall and dietary history questionnaires, while the 3-day or 7-day weighed food diary is rarely used. 

Every instrument has its strengths and limitations depending on the clinical purpose for which they are applied. The main concern about the use of a FFQ is the high risk of bias due to missing data or unreliable completion: Indeed, if the FFQ is applied with the aim to capture the real amount of dietary fiber consumption (in quantity and quality), the number of food items is generally very high, and the time required for the patient to complete the questionnaire could be an obstacle to its correct completion, and this strictly depends on the patient’s food literacy. Second, an FFQ will give clinically reliable nutritional information only if tested and validated for the specific population of the study, thus making it difficult to use the same instrument for different clinical purposes. However, in observational studies, this instrument, when given by a skilled dietician or completed with an expert supervisor, gives clinicians an advantage in evaluating the variety (and related selectivity) of specific food sources of fiber consumed by patients, with the possibility of obtaining a semi-quantitative evaluation without any further characterization other than soluble or insoluble content of fiber.

The nutritional instruments most used in prospective trials, that is, the 24- or 48-h dietary recalls, have the advantage of being more easily used in clinical practice, but the main limitation for dietary fiber is the impossibility in capturing the variety of food sources and thus in evaluating the variation in different types of fiber due to specific food content. 

Certainly, the 3-day or 7-day weighed food diaries are the best instruments to assess the quantity and quality of fiber as well as its variety in food sources, but they are time-consuming both for patients and medical/dietician staff and are more useful for research purposes compared to routine clinical evaluations.

Another emerging unmet need in IBD dietary studies is the characterization of fiber subtypes. As we have previously discussed, the classification of fiber into soluble and insoluble components could help clinicians in particular settings of disease, and it is often the only characterization present in most national and international nutritional databases. However, different food sources contain different types of fiber, which contribute differently to gut health. The increasing knowledge about the importance of the fermentative properties of fiber has led us to reconsider the previous classifications and rethink the characterization of fiber as microbiota-accessible carbohydrates (MACs). Not surprisingly, the interest in most dietary intervention studies in IBD has shifted toward isolated, single fiber or fiber extracts, with the aim of better understanding the fermentable properties of different fibers and the significance of their functionality in the IBD disease course. Nonetheless, parallel efforts should be made to analyze the different contributions of fiber subtypes from specific dietary sources, which could give more realistic information corresponding to real-life consumption of fiber-rich foods as intrinsic fibers.

To date, the most reliable tool is a 23-item short FFQ validated in U.K. population [84] and developed to assess short-term inulin and oligofructose intake through a semi-quantitative evaluation of 23 foods and drinks (8 fruits and vegetables and 15 composite foods, e.g., breads, breakfast cereals, and pastry products). Whelan et al. [34] used the FFQ in a case–control study, demonstrating a lower intake of fructans and oligofructose in active Crohn’s disease compared to inactive CD and healthy controls.

Most recently, a new dietary fiber composition database named FiberTAG repertoire was developed to detail soluble versus insoluble DFs, inulin-like fructans (ITF) (including inulin and FOS), GOS content, and the total DFs content of food products. It is based on the German and Canadian database and in line with the exhaustive literature. Neyrinck and colleagues [85] thus created the new FiberTAG FFQ, a 302-food-item semiquantitative FFQ validated on a healthy Belgian population, which could be a reliable method for measuring long-term (12 months) DFs intake, including prebiotic (oligo) saccharide intakes. The FiberTAG FFQ needs to be tested and validated in other and larger cohorts, but it has the advantage that it can be adapted to specific dietary habits of populations following the aim of specific studies.

The need for a brief assessment tool for the frequency and variety of fruit and vegetable intake drove Ashton et al. [86] to develop the Fruit And Vegetable VAriety (FAVVA) index. 

The authors demonstrated that the FAVVA index is a valid tool to use as a brief indicator of overall fruit and vegetable frequency and variety relative to comprehensive assessment using the Australian Eating Survey (AES) FFQ, suggesting that this instrument could provide a cost-effective and sustainable approach to assessing the frequency and variety of fruit and vegetables intake.

However, different nutritional concerns could rise from this instrument, like the categorization of avocado (a fruit with a higher content of lipids compared to other fruits), potatoes and corn (mainly starchy sources), and legumes (starchy and protein sources) into the group of vegetables (with different macro and micronutrients composition), highlighting the limited application of this tool beyond the simple evaluation of variety in fiber-rich foods. 

Nevertheless, this instrument represents a reliable attempt to go beyond a simple evaluation of the quantity and quality of fiber to better provide rapid feedback on the food variety of fiber sources. 

## 8. Conclusions

Although the paradigm about the dietary intake of fiber in IBD patients has changed in recent years, more efforts are necessary to prompt apply the nutritional indications from scientific guidelines to clinical practice. 

With growing evidence showing the benefits and properties of dietary fiber in IBD, we have provided evidence that highlights the efficacy and tolerance of different dietary interventions focused on dietary fibers in IBD patients in a remission state. However, more well-designed interventional studies in specific clinical settings (active disease, stricturing pattern, and functional symptoms) are urgently needed. 

To provide clinicians and dieticians with useful nutritional information about the ideal quantity, type, and preparation of fiber in the diet for patients with IBD, future RCTs with objective clinical outcomes are needed. Furthermore, the scientific community should continue to advance research on the characterization of fiber subtypes with new methods to evaluate the variety of fiber intake and the overall quality of a diet.

Considering the variable tolerance to different amounts and types of dietary fibers in IBD patients, more preclinical studies linking diet, the microbiota, and SFCA are strongly encouraged. In the future, full comprehension of the mechanisms influencing the variability of response to dietary fiber intake in IBD patients (altered gut microbiome, disease state, and substrate availability for colonic fermentation) will lead to personalized diets developed based on the individual’s intrinsic and extrinsic factors.

While waiting for these scientific advances, healthcare professionals are encouraged to aim to increase dietary fiber intake in the IBD population through a personalized approach.

## Figures and Tables

**Figure 1 nutrients-16-01108-f001:**
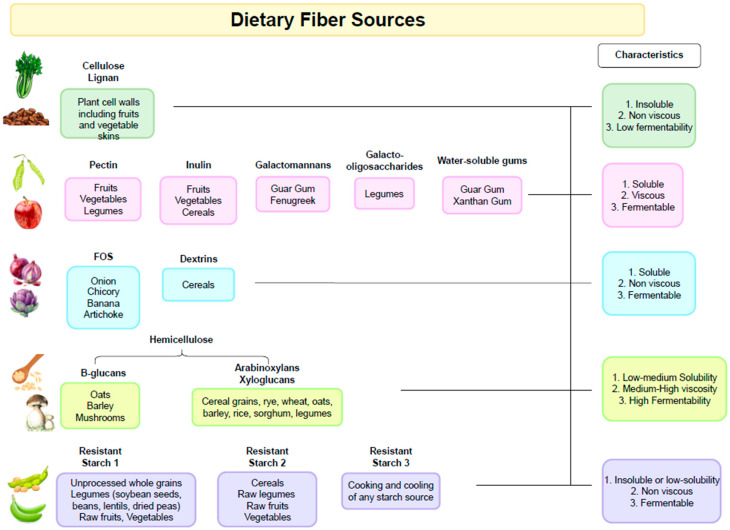
Dietary Fiber Sources.

## Data Availability

Data available in a publicly accessible repository. The data presented in this study are openly available in PubMed.

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
