# Peer review of "Dietary Fiber in Inflammatory Bowel Disease: Are We Ready to Change the Paradigm?"

_nutrients, 2024, doi:10.3390/nu16081108_

Round 1

Reviewer 1 Report

Comments and Suggestions for Authors

Please see attached pdf for comments. 

Comments on the Quality of English Language

Moderate grammar and syntax issues detected that need to be corrected

Author Response

1. Summary

Thank you very much for taking the time to review this manuscript. Please find the detailed responses below and the corresponding corrections highlighted in the re-submitted files.

2. Point-by-point response to Comments and Suggestions for Authors

Comments 1:  Lines 58-78 do not really offer anything new for the topic. I suggest to shorten it or even delete it. In general this whole chapter up to line 124 is not too relevant to your main scope of the review. Consider rewriting.

Response 1:  Thank you for your comment. The entire paragraph 2 has been rewritten. In addition, we eliminated what was not relevant to the main scope of the review.

Comments 2:  I would also add to identify which microorganism is pathogenic agent in IBD

Response 2:  Thank you for your comment. We have added a paragraph describing the impact of dietary fiber on microbiota, acting on harmful and beneficial bacteria in IBD. See line 316 to 331.

3. Response to Comments on the Quality of English Language

Point 1: Moderate grammar and syntax issues detected that need to be corrected

Response 1:  Thank you for your suggestion. We have corrected the grammar mistakes.

Reviewer 2 Report

Comments and Suggestions for Authors

The article concerns the influence of fiber on the course of IBD. unfortunately it is not well written in my opinion. Below I present the comments that should be taken into account

1. the abstract lacks a summary of the research analysis, and I did not find the article search methodology

2. the introduction lacks information about what new information the review introduces compared to others. Analyzing pubmed from the last 5 years alone, there are 11 meta-analyses and over 50 reviews on this topic

3. there are too many repetitions in the article 48%.

should be modified:

first paragraph of the introduction

the first paragraph of section 2 is inadmissible

and all of section 5 and 6

4. important articles on the topic were also omitted

https://www.ncbi.nlm.nih.gov/pmc/articles/PMC8471497/; doi:10.3390/nu13093119

doi:10.5114/pg.2015.52753

https://doi.org/10.1111/apt.17649

The 5th sentence is not understandable to me, line 72

"When derived from a plant origin, dietary fiber may include fractions of lignin 72"

6. information on SCFA, type, changes in proportion and quantity due to intestinal dysbiosis should be added

DOI: 10.1016/j.ijbiomac.2023.126167

doi:10.1007/s10123-022-00309-x.

The 7th conclusion is too general and does not concern remission and exacerbation. there was also no focus on fiber fractions

8. general perception of the article:

the article duplicates content already published

it does not bring any new interesting conclusions

does not take into account a large part of the research conducted on the topic

I recommend the PRISMA guidelines http://www.prisma-statement.org/PRISMAStatement/FlowDiagram

Author Response

1. Summary

Thank you very much for taking the time to review this manuscript. Please find the detailed responses below and the corresponding corrections highlighted in the re-submitted files.

2. Point-by-point response to Comments and Suggestions for Authors

Comments 1: The abstract lacks a summary of the research analysis, and I did not find the article search methodology.

Response 1: Thank you for your comment. As the review is a narrative review, we have performed a comprehensive review of the literature about dietary fiber and IBD and we have chosen to particularly focus on well-designed interventional and observational studies using a food-dietary fiber approach.

To clarify the type of review that we’ve submitted, we’ve specified in the abstract that our work is a narrative review.

Comments 2: The introduction lacks information about what new information the review introduces compared to others. Analyzing pubmed from the last 5 years alone, there are 11 meta-analyses and over 50 reviews on this topic.

Response 2: Thank you for your comment. We’ve revised the introduction detailing the new information presented in our review.

Comments 3: There are too many repetitions in the article 48%. should be modified: first paragraph of the introduction; the first paragraph of section 2 is inadmissible and all of section 5 and 6

Response 3: Thank you for your comment. All paragraphs that you have mentioned have been modified and reformulated.

We’ve rewritten this section 2 to give essential information about dietary fibers that we think are relevant to the main scope of our review.

Comments 4: Important articles on the topic were also omitted:

https://www.ncbi.nlm.nih.gov/pmc/articles/PMC8471497/; doi:10.3390/nu13093119;

doi:10.5114/pg.2015.52753;

https://doi.org/10.1111/apt.17649.

Response 4: Thank you for your comment. All the articles suggested have been added to references and cited in bibliography.

Comments 5: The 5th sentence is not understandable to me, line 72:

"When derived from a plant origin, dietary fiber may include fractions of lignin 72"

Response 5: Thank you for your comment. We’ve decided to rewrite this section to give essential information about dietary fibers that we think are relevant to the main scope of our review.

Comments 6: Informations on SCFA, type, changes in proportion and quantity due to intestinal dysbiosis should be added:

DOI: 10.1016/j.ijbiomac.2023.126167

doi:10.1007/s10123-022-00309-x.

Response 6: Thank you for your comment. This topic, that is not the core of our clinical review, has been added as you’ve suggested in the discussion (line 316 to 332).

Comments 7: The conclusion is too general and does not concern remission and exacerbation. there was also no focus on fiber fractions.

Response 7: Thank you for your comment. All the requested information has been added in the conclusion and a clinical focus on fiber fraction is been implemented in the section 7 (line 379 to 392).

Comments 8: General perception of the article: the article duplicates content already published it does not bring any new interesting conclusions does not take into account a large part of the research conducted on the topic.

I recommend the PRISMA guidelines http://www.prisma-statement.org/PRISMAStatement/FlowDiagram

Response 8: Since the review is a narrative review, the PRISMA guidelines does not applied to our review, we have performed a comprehensive review of the literature about dietary fiber and IBD and we have chosen to particularly focus on well-designed interventional and observational studies using a food-dietary fiber approach.

Round 2

Reviewer 2 Report

Comments and Suggestions for Authors

The authors responded to all the reviewer's suggestions. Thank you for your good cooperation